# Immunological Aspects of the Tumor Microenvironment and Epithelial-Mesenchymal Transition in Gastric Carcinogenesis

**DOI:** 10.3390/ijms21072544

**Published:** 2020-04-06

**Authors:** Jacek Baj, Karolina Brzozowska, Alicja Forma, Amr Maani, Elżbieta Sitarz, Piero Portincasa

**Affiliations:** 1Chair and Department of Anatomy, Medical University of Lublin, 20-090 Lublin, Poland; aforma@onet.pl (A.F.); amrmaanni@gmail.com (A.M.); 2Chair and Department of Forensic Medicine, Medical University of Lublin, 20-090 Lublin, Poland; brzozowskakaro@gmail.com; 3Chair and 1st Department of Psychiatry, Psychotherapy and Early Intervention, Medical University of Lublin, Gluska Street 1, 20-439 Lublin, Poland; elzbietaa.sitarz@gmail.com; 4Clinica Medica “A. Murri”, Department of Biomedical Sciences and Human Oncology, University of Bari Aldo Moro, 70124 Bari, Italy; piero.portincasa@uniba.it

**Keywords:** gastric cancer, epithelial-mesenchymal transition, immunology, immune cells, carcinogenesis, tumor microenvironment

## Abstract

Infection with *Helicobacter pylori,* a Gram-negative, microaerophilic pathogen often results in gastric cancer in a subset of affected individuals. This explains why *H. pylori* is the only bacterium classified as a class I carcinogen by the World Health Organization. Several studies have pinpointed mechanisms by which *H. pylori* alters signaling pathways in the host cell to cause diseases. In this article, the authors have reviewed 234 studies conducted over a span of 18 years (2002–2020). The studies investigated the various mechanisms associated with gastric cancer induction. For the past 1.5 years, researchers have discovered new mechanisms contributing to gastric cancer linked to *H. pylori* etiology. Alongside alteration of the host signaling pathways using oncogenic CagA pathways, *H. pylori* induce DNA damage in the host and alter the methylation of DNA as a means of perturbing downstream signaling. Also, with *H. pylori,* several pathways in the host cell are activated, resulting in epithelial-to-mesenchymal transition (EMT), together with the induction of cell proliferation and survival. Studies have shown that *H. pylori* enhances gastric carcinogenesis via a multifactorial approach. What is intriguing is that most of the targeted mechanisms and pathways appear common with various forms of cancer.

## 1. Introduction

Gastric cancer (GC) is the fourth most common cancer in the world and occupies the second position regarding cancer-related deaths [1,2,3]. This is attributed to the high rate of mortality cases among patients [4,5]. Approximately 990,000 patients are diagnosed with GC each year, of which 738,000 are estimated to die of the malignancy [6]. GC has a higher prevalence among males (approximately two to three times as high) compared to females [7]. The prevalence of GC is also highest in East Asia and Europe, compared to North America and some regions in Africa [8].

The causes of GC are multifactorial. Some causative factors may be decisive in the prognosis of patients and further progression of the malignancy. The most common risk factors and predispositions to GC development include belonging to the male gender, old age, family history of the disorder, smoking, exposure to radiation, obesity, underlying medical disorders or malfunctions (gastroesophageal reflux disease), unhealthy/poor diet, *Helicobacter pylori* (*H. pylori*) infection, genetic factors, or prolonged intake of specific drugs such as non-steroidal anti-inflammatory drugs (NSAIDs) or proton-pump inhibitors (PPIs) [9,10,11,12].

Approximately 95% of all GCs incidents are adenocarcinomas, which may arise either from the gastric mucosa or the most superficial glands of the stomach epithelium [13]. The majority of gastric adenocarcinomas is located in the antrum and body of the stomach and may be classified as diffuse and intestinal carcinomas [14,15]. This classification is based on the Lauren system of classification and depends on its histological features [16,17,18,19,20]. Recently, a molecular classification of GCs was proposed that includes four subtypes of GC–microsatellite instable (MSI), chromosomal instable, Epstein–Barr Virus (EBV) positive and genomically stable [21,22]. Based on recent clinical studies, the prognosis of GC is better among patients who also test positive for EBV within the gastric tumor microenvironment (approximately 9% of all GCs incidents) [23].

*H. pylori* infection makes up one of the most relevant factors regarding the development of non-cardia gastric adenocarcinoma [24,25,26,27]. Because of its invasive properties, *H. pylori* is thought to be responsible for one of the highest rates of chronic infections worldwide. Several virulence factors of *H. pylori* contribute to the stomach invasion and development of GC [28,29,30]. These include cytotoxin-associated gene A (CagA), vacuolating cytotoxin (VacA), or outer inflammatory protein A (OipA) [31]. Furthermore, *H. pylori* infection may either induce or facilitate further epithelial-mesenchymal transition (EMT) process in the GC microenvironment [32].

Various cell types involved in inflammatory processes exist in the GC microenvironment, including macrophages, neutrophils, fibroblasts, regulatory T-cells (T_reg_), B cells, natural killer cells (NK cells), dendritic cells (DCs) or myeloid-derived suppressor cells [26,33,34,35,36,37,38]. All the aforementioned cells along with their products (cytokines) apply to GC development, its initiation, growth, and induction of metastatic properties [39]. Recent research suggests that chronic inflammation and further GC progression is enhanced not only by the EMT process but also by the proinflammatory microenvironment within the malignancy [39]. Inflammation is considered to play a crucial role in tumor initiation growth and increased metastatic properties of tumor cells [40,41]. Regarding the tumor microenvironment, it consists of tumor cells and stroma, a network of blood vessels and a variety of infiltrating inflammatory cells that significantly promote GC progression [42,43]. Besides, *H. pylori* infection significantly alters the microenvironment within the gastric stroma, which may eventually lead to chronic gastritis and further GC development [44,45,46].

## 2. Epithelial-Mesenchymal Transition

EMT refers to the process by which the mesenchymal phenotype is acquired by epithelial cells (ECs) mainly via the reduction of their intracellular adhesions and proliferative capacity (Figure 1.) [47].

Physiologically, EMT can be observed during embryogenesis, tissue development and wound healing [48]. It is also a common phenomenon during carcinogenesis and can either induce or coexist with various malignancies such as breast cancer, thyroid cancer, cholangiocarcinoma, non-small cell lung carcinoma, colorectal cancer, inflammatory bowel disease or GC [49,50,51,52,53,54,55,56,57,58,59,60]. During the EMT process, ECs undergo a series of biochemical reactions that eventually lead to alterations in the cells’ morphology, especially the loss of the polarity of cells [17,61]. The entire cell cytoskeleton undergoes reorganization, cell-cell contacts are gradually impaired, and the shape of the cells changes to a spindle-like, more elongated form [62,63]. During EMT, type 1 cadherin (E-cadherin) is switched to neural cadherin (N-cadherin), which disturbs intracellular binding structures including desmosomes and claudins. Thus, the expression of various epithelial markers such as E-cadherin, cytokeratin, laminin-1, desmoplakin or zona-occludens 1 (ZO-1) is significantly lowered [64,65,66,67]. On the contrary, increased expression of mesenchymal markers (N-cadherin, vimentin, fibronectin, Snail, Slug, TWIST, α-SMA) is observed. EMT enables the acquisition of tumor-initiating cells (TICs) properties by the GC cells along with the development of other malignant features, which is of great importance regarding GC development [68]. Interestingly, EMT is a reversible process—tumor cells that have obtained a mesenchymal phenotype may re-acquire previous epithelial characteristics.

### Definition of Epithelial-Mesenchymal Transition in Gastric Cancer

There is a close relationship between the genesis of GC and EMT, which involves numerous pathomechanisms. Aberrant activation and regulation of GC EMT involves numerous genes, proteins and molecular pathways that are responsible for transcriptional regulation or epigenetic modification. In GC stroma, many cues might influence EMT, among which the *WNT5A* gene seems crucial as it upregulates EMT-related genes [69]. Besides, Fas signaling promotes motility and metastasis in GC in an EMT-dependent manner [70]. GC EMT can be stimulated by Notch activation and p53 deletion [71]. Other mechanisms include GSK3β inhibition, EphA2 overexpression, Wnt/β-catenin signaling activation, aquaporin 3 (*AQP3*) upregulation, overexpression of epidermal growth factor-like domain-containing protein 7 (EGFL7) and *CEACAM6* [72,73,74,75,76]. There is an upregulation of Twist, TGF-β1, Slug, Snail, CD44, and vimentin in dysplasia patients or patients with early GC. On the contrary, E-cadherin is greatly decreased in those patients [77]. Usually, E-cadherin levels in GC tissues have a low expression than that in the normal adjacent tissue [78]. Downregulation of E-cadherin is associated with an invasive plus undifferentiated gastric cell phenotype [79]. Increased expression of *TWIST1* and vimentin in cancer tissues, as well as a decreased expression of programmed cell death factor 4 (*PDCD4*), and E-cadherin expression is all linked with malignant tissue degree [80]. EMT-induced cancer stem cell phenotype contributes to the genesis of GC [81]. In gastric epithelia, the stem cells that lie at the base of the pyloric gastric glands depend on an active Wnt pathway, one that is dynamically regulated, and this is characterized by *Lgr5* expression [82,83]. Activation of EMT in Runx3^-/-^p53^-/-^ gastric epithelial cells is closely followed by the induction of *Lgr5* [84]. The loss of Runx3 in gastric epithelial cells induces EMT, which results in a production of a tumorigenic stem cell-like subpopulation with a high expression of *Lgr5*. Runx3 is deeply involved in the protection of gastric epithelial cells against growth factor signaling and the resulting cellular stemness and plasticity. EMT is linked to cancer stem cells and can induce tumorigenesis and stemness. There is a close association between EMT, GC and chronic inflammation, among which *H. pylori* infection constitutes a major risk factor [6].

## 3. Neutrophils

### 3.1. Tumor-Associated Neutrophils 

Neutrophils are the most abundant leukocyte type that plays a significant role in all immune processes, including cancer progression [85]. Tumor-associated neutrophils (TANs) have been linked to poorer outcomes in various types of cancers [86]. Neutrophils can induce EMT in tumor cells, thereby increasing the migratory ability and invasiveness of GC cells [87]. Not only the elevated absolute peripheral blood neutrophil counts but also the neutrophil/lymphocyte ratio, and neutrophil infiltration in the tumor microenvironment, all make up for poor prognosis [55]. The neutrophil/lymphocyte ratio is a significant prognostic factor in GC progression. High neutrophil production appears to be unfavorable to 5-year survival [88,89]. The score of the absolute postoperative number of neutrophils is an important prognostic factor post-surgery. According to Clausen et al. (2020), TANs density makes up an independent predictor of tumor-specific survival only for women that validate the sexual dimorphism of GC [90].

### 3.2. IL-17a

Neutrophils may promote tumor initiation, progression, and metastasis through the release of several cytokines. IL-17a (a member of the IL-17 family), in particular, is significant in autoimmune diseases and inflammatory processes [91,92]. What makes it particularly interesting is that after binding with its IL-17Ra receptor, IL-17a acts through a novel ACT1-dependent pathway, as opposed to the JAK2-STAT3 pathway triggered by Th1 and Th2 cell families [93,94,95]. The ACT1/TRAF6/NF-κB-dependent signaling is more characteristic to innate immunity and determines the role of IL-17a in autoimmune pathology, neutrophil recruitment and immunity to extracellular pathogens [96]. While IL-17a does not mediate the JAK2-STAT3 pathway directly, it can activate STAT3 indirectly through AKT signaling, it leads to the production of IL-6 triggering the JAK-STAT pathway [97].

The role of TAN-derived IL-17a in GC has been described in a recent study by Li et al. [98]. The study aimed to explore how TANs influence the migration and invasiveness of GC cells. The (indirect) association between TANs and activation of the JAK2/STAT3 pathway was also studied. The study could establish a positive correlation between neutrophils in GC and prognosis. Means of immunohistochemical staining identified neutrophils to be the main cells behind the production of IL-17a. Neutrophil-induced IL-17a promotes EMT through the activation of the JAK2/STAT3 pathway in GC cells, and this effect can be halted via the use of an IL-17a neutralizing antibody. The authors have found an increased number of neutrophils in GC tissues and the number of neutrophils at the invasion margin was strongly correlated to the TNM stage and lymphovascular and perineural invasion. However, no correlation was found between neutrophil count and age, gender, tumor location, tumor size, Lauren type, or histological grade. Finally, Western blot results showed that neutrophils promote the expression of EMT-related markers in GC cells (vimentin, ZEB1, MKN45, and MKN74), while the expression of E-cadherin was significantly decreased. The opposite was found after adding an IL-17a-neutralizing antibody to the co-culture system. The authors propose the use of an IL-17a neutralizing antibody as a novel therapeutic method in GC therapy. Besides, IL-17a gene polymorphism (e.g., G197A polymorphism) is correlated with increased GC risk [99,100].

### 3.3. Bone Marrow Neutrophils

Neutrophil–cancer cell interaction can lead to the polarization of neutrophils towards a pro-tumor phenotype [101]. Zhang et al. (2017) found that bone marrow neutrophils in mice activated by GC cells promote their ability to migrate, thus leading to increased invasiveness. This is made possible through the activation of the ERK pathway and the induction of EMT. The effects were shown to be decreased by inhibiting the ERK pathway. Further studies are necessary to explore the interaction between neutrophils and gastric cell lines and the effects of TAN-induced EMT on GC cell migration and invasiveness. This might provide innovative new strategies for therapy and treatment.

## 4. Fibroblasts

### 4.1. Cancer-Associated Fibroblasts

Fibroblasts are one of the many types of cells present in the GC microenvironment. They can originate from different sources: mesenchymal stem cells (MSCs), EMT cells, and tissue-resident cells [102]. Macrophages interact with MSCs and induce differentiation into cancer-associated fibroblasts; this phenomenon is additionally enhanced by *H. pylori* infection [103,104]. 

Cancer-associated fibroblasts (CAFs) have been shown to have a crucial influence on the initiation, growth, and migration of tumor cells [101]. This is especially due to the reduced cell-to-cell adhesion, enhanced mobility, and constant activation [87,105]. CAFs compared to normal fibroblasts (NFs) exhibit an increased proliferation rate with an activated myofibroblastic phenotype and are a crucial component of the tumor stroma [106]. Some phenotypic markers expressed by CAFs include α-smooth muscle actin (α-SMA), platelet-derived growth factor receptor-β (PDGFRβ), fibroblast activation protein (FAP), fibroblast specific protein-1 (FSP-1,) and vimentin [107,108]. The expression of secreted protein acidic and rich in cysteine (SPARC) in CAFs is associated with a better prognosis [109]. The exact molecular mechanisms of cancer promotion by CAFs are unclear. While it has been shown that CAFs are mainly transformed local normal fibroblasts, the underlying molecular mechanisms of the tumor-stromal fibroblasts activation are yet to be explored with GC [106]. Several studies have explored their role in EMT in particular since EMT enhances the transformation of ECs into fibroblast-like cells in a variety of tissues [107]. The extent of CAF’s prevalence in GC cells was explored in a 2010 study by Zhi et al. [110]. Immunochemistry staining was used to examine the expression of FSP1, α-SMA, and procollagen-1 to determine the prevalence of CAFs and was found to be elevated in tumor specimens compared to normal specimens. The prevalence of CAFs was found to be much higher in metastatic cancers compared to non-metastatic cancers. Still, some studies suggest that CAFs may have a role in inhibiting cancer growth, which further highlights the significance of further research in the field [111].

### 4.2. CAFs and Signaling Pathways

The progression and invasive potential of GC cells are closely associated with the secretion of several cytokines and thus, the activation of various signaling pathways. The cell-to-cell interaction between CAFs and the tumor cells and other surrounding cells including cancer-associated macrophages (CAMs) is essential [112]. In a recent study, CAFs-derived interleukin-33 (IL-33) has been found to mediate the activation of the ERK1/2-SP1-ZEB2 pathway through its receptor ST2L [113]. The secretion of IL-33 by CAFs can be stimulated by TNF-α released by the surrounding GC cells.

CAFs are an ample source of interleukin-6 in the GC environment [114]. This is especially significant in *H. pylori*-associated GC, where the infection induces cyclooxygenase-2 (COX-2)/prostaglandin E2 signaling pathway. PGE2 may induce the hyper-methylation of miR-149 in CAFs, leading to enhanced IL-6 secretion [115]. Physiologically, IL-6 plays a crucial role in immune and inflammatory responses [116,117]. In the tumor microenvironment, it induces EMT via the activation of the JAK2/STAT3 pathway [118]. Wu et al. have shown that both silencing the IL-6 expression in CAFs and inhibiting the JAK2/STAT3 pathway via a specific inhibitor AG490 reduced tumor metastasis in vivo [114]. Similar results were achieved using an IL-6 neutralizing antibody, all posing potential therapeutic targets to diminish the effects of fibroblasts-induced cancer promotion and growth in the GC microenvironment. Co-culturing GC cells with bone-marrow-derived myofibroblasts (BMFs) has shown high expression of IL-6 and hepatocyte growth factor (HGF) secreted by BMFs, which mediated the secretion of TGF-β1 by the surrounding cancer cells [119]. Similar to TANs, CAFs are also a source of IL-17a [120].

Cancer-initiating cells do not exist in isolation but remain in constant interaction with neighboring cells, of which CAFs are of particular significance. CAFs are a source of HGF, TGF-β, VEGF, FGF, and CXCL12 among others [114]. The expression of IL-8, SDF-1, and HGF by CAFs may be induced by an *H. pylori* infection, as-infected fibroblasts have shown an enhanced expression of Snail1 and Twist mRNA [121]. *TWIST1* plays a central role in regulating EMT in the cancer microenvironment, influencing the transition of NFs to CAFs with CXCL2 as the target for transcription [122]. HGF induces the activation of its receptor Met tyrosine kinase (TK) and the HGF/Met activation contributes to oncogenesis and GC progression [123]. Using the HGF-neutralizing antibody has inhibited the transition of normal fibroblasts’ into CAFs in vivo and in vitro [106]. Significantly increased levels of SDF-1 expression were found in GC CAFs in vivo as well [110]. CAFs may also have the capability of ligand-independent activation of EphA2 (erythropoietin-producing hepatocellular protein A2) through the release of VEGF, which aids in EMT progression [124]. Enhanced expression of galectin-1 in CAFs has been linked to an increase in VEGF in the tumor stroma, a crucial factor in angiogenesis promotion (Figure 2.) [125].

A deeper understanding of the role of CAFs in the tumor environment is essential for new-targeted therapy options. CAFs may be a factor in tumor malignancy and developing resistance to 5-fluorouracil [126]. However, several promising potential therapy targets exist in the form of using neutralizing antibodies, JAK2/STAT3 pathway-specific inhibitors, and silencing the expression of pre-cancerous interleukins secreted by CAFs in the tumor microenvironment. Some other options currently discussed include tranilast, which inhibits the TGF-β/Smad pathway and hence EMT, and triptonide, which promotes the production and secretion of tissue inhibitor of metalloproteinase 2 [127,128]. There is a need for further exploration of their role in GC treatment.

### 4.3. Normal Fibroblasts Versus CAFs

There is some controversy on the exact role of normal fibroblasts in the tumor microenvironment. As opposed to CAFs, normal fibroblasts may have a crucial influence on tumor metastasis. In their 2014 study, Xu et al. co-cultured GC cells with dense monolayers of normal fibroblasts at a ratio of 1:10 to mimic the primary environment of cancer metastasis [129]. As a result, some cancer cells morphed to exhibit short, spindle-like morphological characteristics, and increased proliferation and invasion ability. Also, the transformed cancer cells have undergone EMT as shown through the loss of E-cadherin and acquisition of vimentin. Kanzawa et al. have proposed that CAFs stimulate the expression of *WNT5A* in GC cells, which subsequently maintains cancer stem cell properties and induces EMT [130]. Through interaction with tumor stromal cells, CAFs acts as an important modifier of cancer promotion, progression, and metastasis. Normal, non-cancerous fibroblasts have been shown to undergo phenotypic change and exhibit CAF-like traits when grown in vivo in the GC microenvironment [131]. Fibroblast activation protein (FAP) is expressed selectively by CAFs and pericytes in over 90% of human epithelial cancers, including GC [110]. In a 2009 animal study, Santos et al. showed that FAP-targeted therapy inhibits tumor stroma genesis [132]. FAP has been shown to promote EMT via the Wnt/β-catenin signaling pathway, thus facilitating cancer invasiveness and progression [133].

### 4.4. Fibroblasts and Tumor Metastasis

CAFs play a crucial role in the tumor’s metastatic potential [134]. They stimulate an epigenetic change in microRNA (miR)-200, reducing its expression and thus promoting cancer invasion and peritoneal dissemination [135]. CAFs themselves show downregulation of miRNA-214 as compared to NFs, which mediates the upregulation of vimentin and thus EMT [136]. However, the ability of tumor cells to migrate and live in healthy tissues may not automatically correspond to the cells’ ability to form a metastatic tumor mass. The “traveling tumor cells” hypothesis suggests that cancerous cells may disperse in tissues without forming a metastasis, a notion separate from the circulation of tumor cells and the disseminated tumor cells [137]. Tumor cells that underwent EMT and thus acquired enhanced invasiveness, may then need to undergo the reverse process of mesenchymal-epithelial transition (MET) to form a metastasis away from the primary tumor source. In the environment where fibroblasts underwent MET to gain an epithelial phenotype, a significantly increased amount of suppressor of cancer cell invasion (SCAI) was observed [121]. However, the exact role of SCAI in cancer progression is unclear.

One of the most preferable GC metastasis locations is the liver, which is usually associated with a poor treatment outlook. Also, in a recent study, a close correlation has been shown between the presence of α-smooth muscle actin (α-SMA)-positive CAFs in the metastatic stroma and poor patient prognosis [138]. Upon stimulation by TGF-B in the tumor microenvironment, these CAFs were found to secrete lysyl oxidase (LOX), which facilitates tumor growth and progression.

## 5. Macrophages

### 5.1. Tumor-Associated Macrophages: M1 and M2 Phenotypes

Macrophages are among a variety of immune cells present in the GC microenvironment. Their plasticity enables them to play a crucial role in the tumor microenvironment [101,139]. Macrophages can acquire either an M1 (classical) or M2 (alternative) phenotype [85]. Tumor-associated macrophages (TAMs) undergo the M2 activation, a feature essential to their role in tumor growth and progression. The relationship between cancer cells and macrophages is two-fold: cancer cells induce the M2 phenotype differentiation in macrophages, which promotes tumor growth and progression [140]. An M2 predominance was associated with worse overall patient survival [141]. The expression of the PD-1 receptor on M2 macrophages allows for immune escape in cancer cells through interaction with its PD-L1 receptor; also, M2 macrophages can neutralize the effects of pro-inflammatory, anti-tumor M1 macrophages. The secretion of Il-6 and Il-8 by GC-derived mesenchymal stromal cells and further activation of the JAK2/STAT3 pathway induces the polarization of macrophages into pro-tumor M2 phenotype, which consequently promotes GC metastasis via advancing the EMT process [142]. Likewise, GC metastasis is promoted by matrix metallopeptidase 9 (MMP-9) via the activation of PI3K/AKT/Snail signaling pathway and subsequently induced EMT [143,144]. The M2 macrophages mediate a Th2 immunological response, while classical M1 cells promote cancer regression through the Th1 response pathway. Najafi et al. suggest that therapy targeted at phenotype reversal towards the classical M1 phenotype in macrophages and blocking the PD-1/PD-1L pathway may be a novel therapy [140,145].

### 5.2. Tumor-Associated Macrophages in Gastric Cancer

In GC patients, high infiltration of TAMs in the tumor microenvironment is associated with poor prognosis and more malignant phenotypes with increased angiogenesis [146,147,148,149,150,151,152]. However, the role of M2 macrophages in EMT is unclear. In GC tissues (*H. pylori*-positive), tumor-infiltrating macrophages express activated mesenchymal-epithelial transition factor (MET) [153]. A 2017 study by Fu et al. has shown a significant correlation between the expression of the CD68 marker characteristic to TAMs and losing expression of the epithelial marker E-cadherin, as well the presence of the mesenchymal marker vimentin [146]. Zhang et al. (2017) proposed increased TAMs infiltration and hypoxia-inducible factor-1 alpha (HIF-1α) as potential factors stimulating the induction of EMT [154]. Likewise, TAMs induce the expression of forkhead box Q1 (*FOXQ1*) promoting EMT with invasion and migration properties [155]. However, the exact mechanism of EMT promotion by TAMs is not fully understood.

TAMs can promote the growth of GC, promotion, and infiltration through several signaling pathways. TAMs are linked with the activation of the β-catenin pathway, one factor promoting cancer cell invasiveness in GC [156,157]. This means that the suppression of the β-catenin pathway could be a possible treatment for GC [158]. M2 macrophages have also been linked to the impairment of NK-cell function via TGF-β. A decreased amount of NK cells has been found in the GC microenvironment with ample TAMs present (Figure 3.) [159].

TAMs produce several factors promoting angiogenesis and lymphangiogenesis in GC [106,160,161]. An increased presence of macrophages in the tumor microenvironment is associated with a higher expression of vascular endothelial growth factor (VEGF) and VEGF-C. Macrophage migration inhibitory factor (MIF) makes up an essential link between chronic inflammation and further gastric carcinogenesis due to the promotion of EMT and cellular transformation [162]. Various interleukins expressed by TAMs are also associated with tumor progression and prognosis, including IL-6, IL-10, CCL5, and CXCL8. IL-6 present in the tumor microenvironment induces the M2 phenotype differentiation in macrophages through the activation of the STAT3 pathway [146]. IL-6 secreted by intraperitoneal TAMs promotes peritoneal dissemination of GC [163]. CXCL-8 present in the GC stroma is secreted mostly by macrophages, leading to an increased expression of PD-L1 on their surface [164]. This reduces CD8+ T-cells infiltration and impairs the anti-tumor immune response. Macrophages are a significant source of IL-10 in GC. The levels of IL-10 are significantly increased in both the tumor tissue and serum of GC patients. IL-10 has been found to influence GC invasiveness and migratory ability in vitro through the activation of the c-MET/JAK3 pathway [165]. Finally, there is a positive correlation between high expression of CCL5 and CD68 in GC cells. CCL5 promotes cancer growth and proliferation and may serve as another potential therapeutic target [166]. It was observed that docking protein-1 (*DOK1*) positivity in stromal cells correlates with increased levels of nitric oxide synthase in CD68+ TAMs and thus, poor prognosis [167]. Besides, legumain (tumor-promoting protein) in TAMs is suggested to play a crucial role in the proliferation and invasion of GC cells [168].

Since the level of infiltration of TAMs has a direct correlation with survival outcomes, it can serve as an independent prognostic factor in GC [146]. TAMs have been used as biomarkers in many cancers to date and have shown great potential in serving as prognostic biomarkers in GC as well [169]. Potential TAM-targeted therapy could be another treatment option for GC patients [170].

## 6. Interleukin-23 and STAT3 Pathway

### 6.1. Interleukin-23

Interleukin-23 (IL-23) is a heterodimeric cytokine, which belongs to the IL-12 cytokine family [171]. Over the last decade, a significant role was assigned to IL-23 in terms of the induction and progression of inflammation and inflammatory diseases. A heterodimeric IL-23 is produced by the interaction of IL23A and IL12B. The entire process is controlled by the expression of the IL23A gene, which is over-expressed in cases of the inflammation of CagA-positive *H. pylori* [172]. IL-23 is produced mainly by DCs and myeloid cells via toll-like receptor (TLR) through either exogenous or endogenous signals including damage-associated molecular patterns (DAMPs), pathogen-associated molecular patterns (PAMPs) or tumor-associated factors [173,174,175]. IL-23 has a high affinity to the IL-23R presented on various immune cells including T cells, monocytes, DCs or NK cells [176,177]. Because of the induction of inflammation and angiogenesis in tissues, IL-23 is widely involved in the progression of carcinogenesis. The mechanism of IL-23 action involves the induction of proliferation and expression of Th1 and Th17 cells. It suppresses the antitumor properties of T cells and the antimetastatic functions of NK cells. It was observed that deficiency of IL-23 equals the lower susceptibility of tumor development and metastasis [178]. Particularly, the IL-12/IL-23 axis of inflammation is the potential target for cancer therapy [175,179].

### 6.2. IL-23 and Carcinogenesis

Tumorigenesis is preceded via the activation of several pathways including NF-κB, cGAS-sTING, and STAT3 pathways in particular [180,181,182,183,184]. *STAT3* is expressed in a vast number of cells, being responsible for the control of the cellular processes of proliferation, differentiation, survival, and apoptosis. The over-activation of IL-23 depends on the STAT3 pathway activated by the phosphorylation via tyrosine kinase 2 and Janus kinase 2 [185,186,187]. An elevated level of IL-23 correlates with the poor clinical outcome and prognosis of cancer-dependent patients [188,189]. The prolonged over-activation of the STAT3 pathway, particularly in malignant cells, controls the survival and proliferation rates via Bcl-X_L_, Bcl-2, Mcl1, *c-Myc*, Survivin, or Cyclin D1 levels [190,191,192,193]. *STAT3* induces the transcription of various oncoproteins and transcription factors and IL-6/JAKs, EGFR or Src in a process of dysregulation and further deletion of regulatory proteins [194].

IL-23 is involved in the induction of the STAT3 pathway and EMT within gastric cells [195]. IL-23 is mainly secreted with other interleukins including IL-1β, IL-6, IL-10, IL-12p40 or IL-12p70 by macrophages; IL-23 is also released by T cells and GC cells [196]. Particularly, IL-17 and IL-23 are highly abundant in cases of advanced gastritis and GC incidents related to *H. pylori* infection [197,198,199,200]. It was reported that the expression of IL-23 receptor (IL-23R) is much higher in GC tissues compared to adjacent, not affected tissues. Also, the level of IL-23 is increased in the serum of patients with GC [201]. The attachment of IL-23 with its receptor intensifies the migration and invasion properties of GC cells mainly by the induction of the STAT3 pathway and further EMT processes. The proliferation rate of cells is independent of the IL-23 level. The higher quantity of IL-23R within GC tissues is associated with larger tumor size and worse clinical outcomes. The infection of VacA-positive *H. pylori* strains results in the increased number of Th17 cells in the lamina propria of gastric mucosa, which depends on the IL-23 level according to the mice model [202]. Also, susceptibility to GC may be enhanced by micro-RNA binding site single-nucleotide polymorphisms (SNPs) within the IL-23/IL-17 pathway [203]. Regarding the polymorphisms within IL-23R specifically, two genotypes, rs1884444 and rs6682925, are associated with the poorer prognosis of GC patients [204]. Increased level of IL-23 stimulates the further release of IL-17 via activation of IL-17A/IL-17RA/NF-κB signaling within the GC microenvironment. Since IL-23 levels are also detectable in the serum of affected patients, it was hypothesized that IL-23A concentration may make up a potential indicator of poor prognosis of GC [205]. Regarding the blood biomarkers of GC, disintegrin and metalloproteinase-8 are considered being useful, since their levels are highly correlated with IL-23 levels in blood [206]. There is a correlation between IL-23 level and many mesenchymal and embryonic stem cells in patients with GC [207,208].

## 7. Mesenchymal Stem Cell-Derived IL-15

Mesenchymal stem cells within the tumor microenvironment are reprogrammed to a tumor-specific pattern eventually giving rise to GC and EMT progression [209,210,211]. MSCs are agitated to migrate towards tumor tissues via various factors excreted by tumor cells, including SDF-1, IL-6 or PDFG. Besides migration properties, MSCs develop into tumor-associated MSCs (TA-MSCs) and CAFs [212]. Interestingly, MSCs stimulate angiogenesis because of the secretion of VEGF, PDGF, basic fibroblast growth factor (bFGF), and angiopoietin-1, providing tumor immune escape via the release of TGF-β at the same time [213,214,215]. The secretion of TGF-β and SDF-1a mediates the recruitment of CAFs to the tumor microenvironment. The presence of MSCs within tumor microenvironment induces tumor characteristics, the increased metastatic potential of cancer cells and drug resistance. MSCs also promote the proliferation of cancer stem cells (CSCs) within the tumor environment. Secretion of CXCL1, CXCL5, 6 and 7, IL4, IL8, IL10, IL17b, S100A4, and EGF stimulates cancer cells. The differentiation of MSCs into CAFs through the CXCL12/CXCR4 axis induces the EMT process [216].

Recently, the role IL-15 derived from GC MSCs has been investigated in terms of EMT induction and further GC progression. It was reported that IL-15 levels are increased in both serum and tissues of GC patients comparing to control groups. IL-15 is a pleiotropic cytokine that belongs to the 4-a-helical bundle family of cytokines and acts via the attachment to the receptor complex comprising IL15Rα (CD215), IL2Rβ (CD122) and the common γ (CD132) chains [217]. IL-15 is involved in the control of development, homeostasis, and functions of T cells, NK cells and NK-T cells [218,219]. It is also a vital cytokine for the control of B cells, DCs, macrophages, and mast cell functions. IL-15 acts as a growth factor for both immune and tumor cells. IL-15 derived from GC MSCs enhances the migration properties of gastric cells, influencing the progression of the EMT process [220]. Over-activation of STAT3 and STAT5 pathways also depends on Il-15 levels. GC MSCs-derived IL-15 increases the number of T_reg_ cells via the activation STAT5 pathway in CD4+Foxp3+T cells. An elevated number of Foxp3+T_regs_ is associated with enhanced tumor metastasis in various cancers. Further, an increased quantity of Foxp3 in CD4+T cells results in an elevated percentage of PD-1 in T_regs_.

## 8. Lymphocytes

The GC microenvironment is characterized by the enriched infiltration of tumor lymphocytes. These stimulate the inflammation processes within the gastric microenvironment via the secretion of various molecules and mediators such as cytokines, chemokines and matrix metalloproteinases. The presence of CD8+ cytotoxic T cells correlates with improved prognosis of GC patients since CD8+ lymphocytes promote antitumor effects [221,222]. Furthermore, CD8+ lymphocytes stimulate the release of proinflammatory Il-17, promoting tumorigenesis and worsening the clinical outcome of GC [223]. Th1, Th2, Treg, and Th17 are all inducers of EMT and can be obtained due to the differentiation of naive CD4+ T helper cells [224]. A high Th1/Th2 ratio correlates with a better prognosis of GC incidents. Likewise, the higher infiltration of Foxp3+ T_regs_ equals to poorer prognosis and higher metastasis rates [225]. Further, GC tissue-derived MSCs induce the differentiation of CD4+ cells into T_reg_ cells (but not Th17 cells) [226].

It was reported that PD1^+^ peripheral CD8^+^ T-cells might make up an independent prognostic marker of severity of GC [227]. The downregulation of microRNA-451 (miR-451) in GC tissues is associated with the increased infiltration of T cells. Subsequently, the increased amount of T cells results in the further enhanced differentiation of Th17 cells [228]. It was observed that CD4^+^ FOXP3^+^T cells within GC tissue are more abundant than CD4^+^ FOXP3^-^ T cells. CD4^+^ CD25^high^ T cells present higher expression of 10/IFN- γ gene and lowered TGF- β levels [229]. Li et al. (2019) observed that smaller tumor size correlates with higher expression of peripheral CD4+ and CD8+ cells [230].

Studies have shown that dense tumor-infiltrating lymphocytes (TIL) are associated with improved prognosis in certain immunogenic tumors, and GC is believed to be an immunogenic tumor [231]. High expression of pan-T cells or TILS in tumor tissue had a significant correlation with favorable overall cancer survival. The implication is that the adaptive immunity that’s mediated by T lymphocytes may serve as a potent anti-tumor response via eradication of cancer cells and avoiding tumor growth [232]. In a study involving 200 patients with GC, the CD3, CD8, and CD45RO high-density groups had prolonged survival times compared to the corresponding groups (low-density) [233]. Recently, Lee et al. conducted a study in vitro, which showed that adaptive immune responses might be initiated in the inflammatory microenvironment of gastric tumors, with TILS being able to induce apoptosis in GC models [234]. A release of tumor antigens into the tumor microenvironment via radiotherapy or cytotoxic chemotherapy induces cell-mediated apoptosis through the activation of cytotoxic T-cell lymphocytes.

## 9. Conclusions

While we understand that *H. pylori* was identified as a carcinogen over 20 years ago, medical researchers are still trying to unravel the novel oncogenic mechanisms utilized by this pathogen. *H. pylori* has great influence over several host pathways to trigger the development of cancer. Although not all the mechanisms are fully understood, evidence suggests that as *H. pylori* triggers the infection, the response from the host results in the creation of an environment that encourages the development of tumors. Changes are undergone by the cell in response to *H. pylori* infection involving several hallmarks of cancer progression. These include DNA methylation and damage, EMT pathway activation, suppression of tumor suppressors, activation of anti-apoptotic effectors, and pro-proliferative effectors. It is worth noting that *H. pylori* has the potential to target cancer pathways as a means of inducing gastric carcinogenesis.

## Figures and Tables

**Figure 1 ijms-21-02544-f001:**
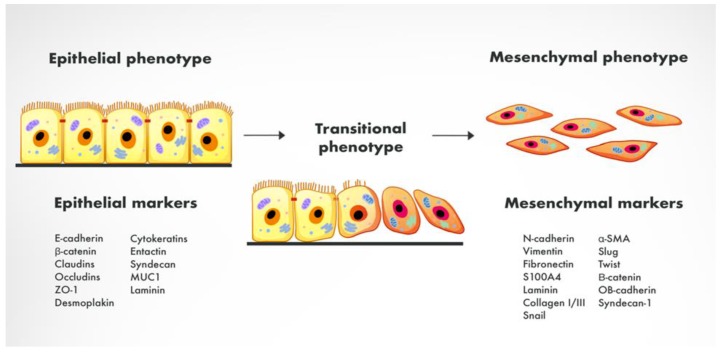
The process of epithelial-mesenchymal transition with specific epithelial and mesenchymal markers.

**Figure 2 ijms-21-02544-f002:**
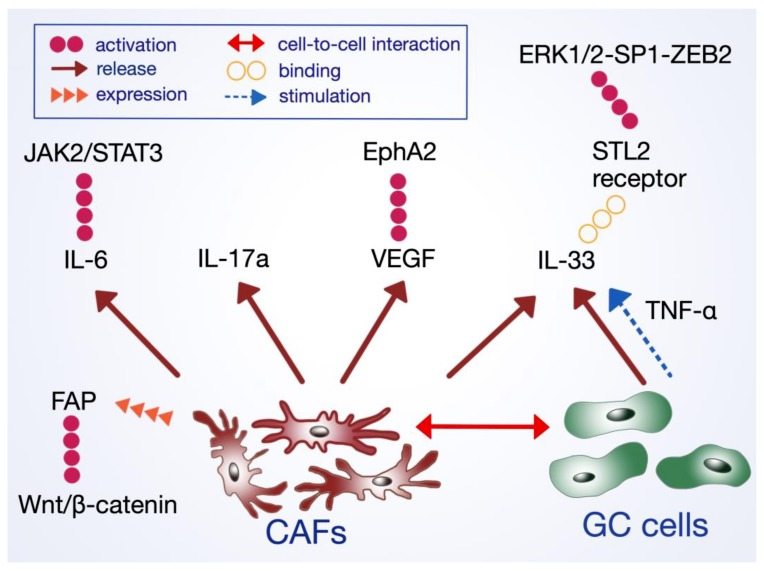
Mechanisms of cancer-associated fibroblasts action in EMT and gastric cancer progression.

**Figure 3 ijms-21-02544-f003:**
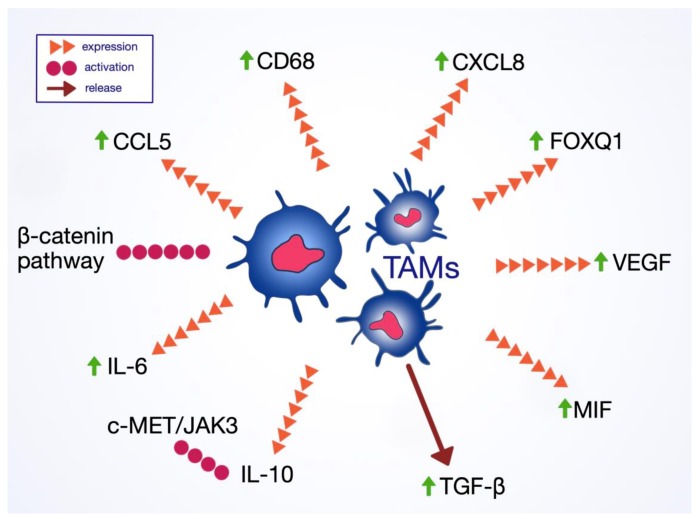
Involvement of tumor-associated macrophages in EMT and gastric cancer progression.

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
