# Peer review of "Immunological Aspects of the Tumor Microenvironment and Epithelial-Mesenchymal Transition in Gastric Carcinogenesis"

_ijms, 2020, doi:10.3390/ijms21072544_

Round 1
Reviewer 1 Report
Thank you for submitting a manuscript on this important topic. The authors provide a comprehensive review of the molecular underpinnings of the tumor microenvironment in gastric cancer.
At times the authors discuss other conditions such as lung cancer, arthritis, and other conditions that are not relevant to gastric cancer and H. pylori. The authors should omit any extraneous information regarding other disease processes that are irrelevant to the primary focus of the review in order to make the manuscript more concise.
There are also errors in some references. For instance, references 80 and 81 lack titles. Reference 187 includes text of the manuscript. The authors should carefully go over their references to ensure that they are accurate.
It would be helpful if the authors could generate a figure that incorporates several concepts included in their manuscript such as tumor-associated neutrophils, IL-23, and fibroblasts.
Author Response
Jacek Baj
Medical University of Lublin
Department of Human Anatomy
Jaczewskiego 4
20-090 Lublin, Poland
jacek.baj@umlub.pl
30th March 2020
Dear Reviewer,
Thank you very much for reviewing our manuscript. We appreciate the interest and commitment you have provided for this work. We are very grateful for your extremely precious comments. We are convinced that thanks to your suggestions this manuscript will be much more valuable.
We are pleased to submit explanations and details of our revisions in the manuscript entitled ‘Immunological aspects of the tumor microenvironment and epithelial-mesenchymal transition in gastric carcinogenesis’.
The followings are our point-by-point responses:
- We have deleted several unnecessary information about other conditions throughout the whole text, which were not related to gastric cancer and irrelevant regarding the topic of the manuscript.
- We have added titles in references number 80 and 81. We have deleted the text from a reference number 187. We have also gone through other references to ensure whether they are provided in a proper form.
- We have also added two additional pictures (which are original figures) in the manuscript. Both of them visualize mechanisms of tumor-associated macrophages and cancer-associated fibroblasts in EMT and gastric cancer progression.
- We have also made several English corrections in the whole manuscript.
We hope that after this revision, the manuscript is of a higher quality and worth reading.
We wish you all the best
Sincerely,
Jacek Baj,
on behalf of the authors
Reviewer 2 Report
It is a comprehensive review.
The title includes EMT, but definition of EMT in gastric cancer is not specified. Usually loss of E-cadherin does not mean EMT in practice and vimentin is very rarely positive in GC. Just add the issue on this point. Undifferentiated or non-cohesive type is not necessarily the phenotype of EMT.
Author Response
Jacek Baj
Medical University of Lublin
Department of Human Anatomy
Jaczewskiego 4
20-090 Lublin, Poland
jacek.baj@umlub.pl
30th March 2020
Dear Reviewer,
Thank you very much for reviewing our manuscript once again. We are very grateful for the commitment you have provided during the revision of this manuscript and appreciate your precious comments.
We are pleased to submit the manuscript entitled ‘Immunological aspects of the tumor microenvironment and epithelial-mesenchymal transition in gastric carcinogenesis’.
The manuscript has been checked once again in terms of English by one of our native speaker friends and we hope that now it is much more improved.
We have added an additional subparagraph entitled ‘Definition of epithelial-mesenchymal transition in gastric cancer’ and hope that this has improved the overall EMT description regarding gastric cancer.
Sincerely,
Jacek Baj,
on behalf of the authors
Round 2
Reviewer 2 Report
The addition of the definition of EMT improved the manuscript.